



# Enjoying the ice. Dutch Winter landscapes, weather and climate in the Golden Age, 17th century

Alexis Metzger[1]

[1]Institute of Geography and Durability, University of Lausanne, Lausanne, Switzerland

*Correspondence to*: Alexis Metzger (alexis.metzger@unil.ch)

**Abstract.** This article explores Dutch winter landscapes from the 17th century, in light of written climatic sources. It investigates different kinds of climatic elements and winter weather types that were favoured by the artists during the Little Ice Age. The comparison between a corpus of paintings and narrative records show an overrepresentation of cold and dry weather in painted representations, in comparison to the written documents. Indeed, we can estimate that such particularly cold and dry weather corresponded to less than 20% of winter days. Thus, the Dutch painters produced a winter imagery supported by icy scenes, in which the Dutch practiced skating. We interpret this choice by examining hypotheses based around four themes: climatic, religious, political, and social. Finally, despite their historical relevance, these winter landscapes are a *genre*, and only show very partially the diversity of winter weather during the 17th century and the Little Ice Age.

**Keywords:** Weather, climate, winter, Little Ice Age, Golden Age, painting, landscape

## 1. Introduction

During the 17th century, the Dutch population experienced the Little Ice Age (LIA), "an epoch of cooler average temperature prevailing generally from the end of "medieval warming" to the beginning of our contemporary era of global warming" (de Vries, 2013, p.371). Cold winters and fresh summers were frequent. The duration of this period is not precisely defined (Büntgen and Hellmann, 2013; Le Roy Ladurie, 2004; White, 2013). Nevertheless, the authors all agree that the 17th century was one of the coldest centuries in the second millennium (Le Roy Ladurie, 2004; Mauelshagen, 2012). Numerous researchers have been working on this period, trying to estimate its "natural" components, such as temperature and precipitation. Another research focus has been on the societal impacts of this colder period. Yet, the link between a climatic trend and the societies' reaction and adaptation to it, are still being discussed (de Vries, 2013; White, 2013).

In parallel of this climatic variability, "seventeenth-century Dutch painting is the most obvious manifestation of all the changes that took place during the Golden Age" (North, 1999, p.1). Through the 17th century, landscapes became a "genre" (Alpers, 2009; Sutton et al., 1987; Todorov, 2017). Included in this new artistic category, winter landscapes were produced by numerous painters, among which the most famous is probably Hendrick Avercamp. He was the first Dutch painter to represent winter scenes at the beginning of the 17th century, after the Flemish – Brueghel the older, Jacob Grimmer, or Luckas van Valckenborch. Avercamp's winter scenes were appreciated and rediscovered in the context of great exhibitions (Roelofs, 2009; Suchtelen, 2002). Many other Dutch painters are also well-known for their winter landscapes, such as Jan van Goyen, Aert van der Neer, or Jacob van Ruisdael.



In this paper, we focus on a period going from 1608 to 1672. The 1608 winter was severe, and also corresponds
to the first painting of a Dutch winter landscape, by Avercamp. In 1672, the *rampjaar* disaster took place, as France
governed by Louis XIV invaded the Netherlands. The Golden Age thus came to an end (Prak, 2009).
Our research connects the two fields of history of art and history of climate together, along with the study of
cultural appropriation, and depictions of weather and climate (Hulme, 2016; Jankovic, 2000). It is nowadays a
fructuous place for interdisciplinary research between geography, anthropology, ethnology, and history. Several
studies pertaining to the LIA emphasized links between the climate and cultures (Behringer, 1999; Behringer et
al., 2005). Yet, "ordinary" landscape paintings have not been closely examined. Only a few studies have explored
European landscape paintings with climatic analyses (Camuffo, 1987, 2010; Gedzelman, 1991; Nussbaumer,
2012; Zerefos et al., 2007). Indeed, most geographers looking at paintings either draw on cultural geography, or
visual studies (Daniels and Cosgrove, 1988; Grison, 2002; Staszak, 2003). Though such areas of research are
important in our own interpretations, natural – including climatic – elements are still often neglected, whereas
"like photographs, paintings constitute a form of evidence that can provide information (albeit mediated) about
past landscapes that is different from the information contained in other sources" (Gaynor and McLean, 2008,
p.189). Through this pluri-disciplinary focus, we investigate Dutch winter landscape reflections, and likelihood to
winter weather at that time. To make this comparison, we use the methods of climate historians, art historians, and
cultural geographers altogether. In our view, it is necessary to devote greater attention to the paintings displaying
both historic and climatic dimensions. Do these winter paintings really show the natural and social conditions of
winter at that time? This question has only been partially encompassed (Burroughs, 1981; Degroot, 2014; Goedde,
2005; Robinson, 2005; Roelofs, 2009). We do not only focus on what is depicted in the paintings, as art historians
do, but also on what could have been depicted according to the written sources. What is missing is probably not a
coincidence, but a choice requiring explanations.
First, we highlight the corpus of paintings, and the written sources we use. Then, we show how the comparison
between the paintings and the archival documents reveals the kind of weather the artists preferred. We finally
suggest several hypotheses explaining these choices.
**2. Material and methods**
**2.1. A corpus of winter landscapes**
The analysis is based on a corpus of 49 winter paintings. The total amount of winter landscapes produced during
the Golden Age is unknown. Many paintings have either disappeared, or are kept in private collections. Therefore,
it is impossible to be exhaustive in our findings. In order to be representative, we have decided to focus on the 49
paintings reproduced in the only international exhibition devoted to winter landscape paintings in the 17th century
(Suchtelen, 2002). Following this recension, we proceeded to a statistical analysis inspired by researchers in
quantitative geography and history of art (Brulez, 1986; Gedzelman, 1990; Joyeux-Prunel, 2008, 2013). The idea
is to understand whether or not an element identified in one painting often appears in the other paintings as well.
Thus, we are not only able to examine the weather in one specific painting, but also to get a grasp of the main



weather types painted in our entire corpus. In a way, it creates an "average" of the weather elements, and allows
us to picture the "climate" in one artificial image.
One landscape painting shows a particular moment in a day. Hence, the image exposes one weather type, defined
by the following criteria: the rain or the absence of rain, where the wind comes from, the type of clouds painted,
etc. A few authors tried to analyse landscape paintings in the light of meteorological processes (Gedzelman, 1991).
As for the 17th century, they concluded the weather is, in majority, coherent with the scientific knowledge we have
of that period nowadays. Even if the "real" skies might happen to be exactly those painted at the time, meteorology
as a science was nevertheless at its very first attempts. For instance, cloud classification was only invented in 1802.
Even more impressive, as the painters used to work indoors, they could see the sky from the inside, but not the
scene they wanted to create.

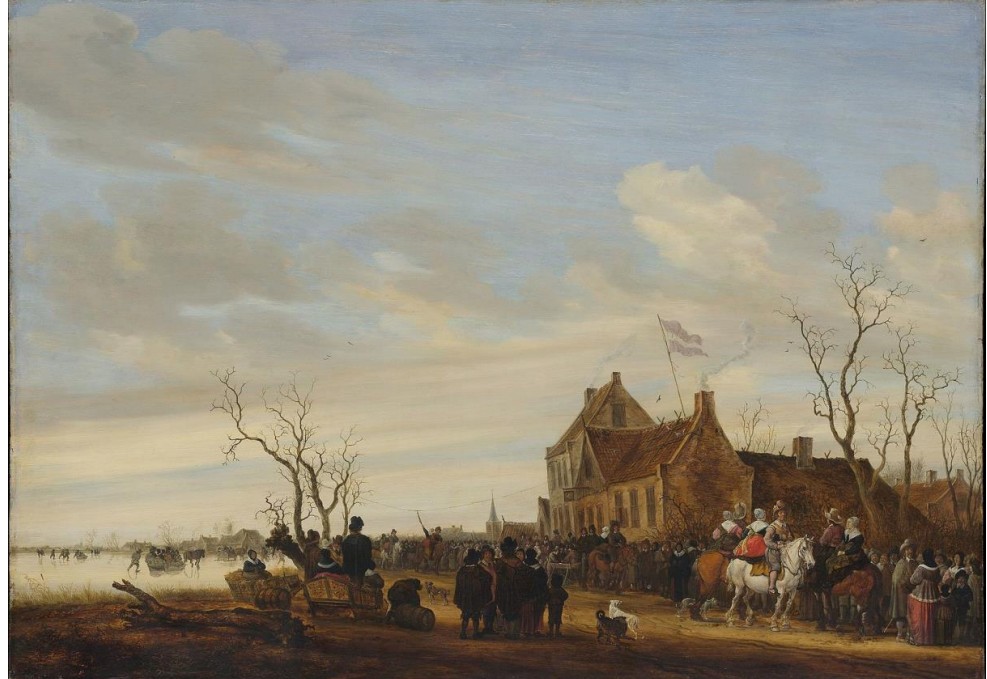


Figure 1: Salomon van Rusydael, Drawing the Eel, early 1650s, Metropolitain Museum of Art, oil on panel, 74,9
x 106 cm
https://commons.wikimedia.org/wiki/File:Drawing_the_Eel_MET_DP147902.jpg

In this painting by Jacob van Ruysdael (Fig. 1), there are some elements with meteorological coherence. On the
left side, one distinguishes icy water. A few people and horses are moving toward the inside, which means the tide
is reasonable, min. 15 cm. On the right side, the trees have no leaves. The leaves have already fallen, probably
because of the wind and tempests, which are quite frequent in the Netherlands during winter. As the trees have no
leaves, the scene certainly takes place after November, and probably after December. The flag, at the top of the
big house, indicates the wind is coming from the left. Moreover, the people's shadows indicate the sun is shining



from the left. Finally, the number of people watching the game tells us the scene takes place during the afternoon.
These elements all show the wind comes from the left side, from the west. The clouds are totally coherent: cirrus
in the top and right side, cumulus in the middle, and probably stratus in the left and bottom side of the painting.
After a period of severe frost, caused by the stagnation of an anticyclone -as testified by the absence of snow-, a
perturbation comes from the west, and will soon cover the sky. In a few hours, it will probably snow. Thus, as for
the majority of paintings we focus on, all the elements are well painted, according to a *weathered* "reality" (in
Mike Hulme's terminology). Yet, using this kind of painting, is it possible to have any *climatic* indication?
**2.2.    Climatic sources**
Our study includes written sources as well, since "most reconstructions from archives natural cannot be broken
down to seasonal or monthly resolution and they do not always yield a distinct separation of the effects of
temperature and precipitation" (Brázdil et al., 2005, p.375). Winter narratives rely on climatic archives, often
disseminated among the different archive categories (agriculture, economy, war, religion, etc.). Indeed, throughout
history, people have always given their impression about the weather (Pfister et al., 1999; Thomasset and Ducos,
1998). Nowadays, even though the scientific, meteorological measures are well developed around the world, many
people still share their own observations (Endfield and Morris, 2012). Martin de la Soudière speaks about
"metéophiles", or weather enthusiasts (de La Soudière and Tabeaud, 2009).
In the 17th century, weather enthusiasts were, in a large majority, men. They were either members of the clergy
and nobility, or scientists, farmers, doctors, and so on. David Fabricius is one of the most interesting of them. He
lived in Friesland. He had a very accurate perception of the weather; for instance, he used around 70 expressions
to speak about frost (Metzger and Tabeaud, 2017). At the beginning of the 17th century, savants, such as Nicolaes
van Wassener, used the first thermometers. From November 1612 to March 1615, Isaac Beeckman set up the first
"weather station" in Amsterdam (featuring a weather vane and a thermoscope, an ancestor of the thermometer).
From 1655 to 1667, Fabio Chigi – pope Alexander VII – also made meteorological observations. At the end of the
17th century, a shipowner and merchant, based in Koog aan de Zaan, recorded the weather over a 60-year period.
To have a good overview of past weathers described by weather enthusiasts, we chose to use Jan Buisman's books
(Buisman, 2011; Buisman and Engelen, 2000, 2006). The author has produced an impressive and critical inventory
of different, primary and secondary sources: chronicles, letters, weather observations, etc. This corpus provides a
relevant overview of winters during the Golden Age.
**3.   Results**
**3.1.    Paintings and climatic diachronies**
We surveyed our corpus of landscape artworks and show that the climatic elements were chosen by the painters.
They represented ice, but no rain. Where there is snow, it never falls, and is always very thin on some parts of the
rooftops. The preferred clouds are the stratus and the cumulus. The skies are always depicted with clouds, which



coheres with today's observations. From the west to the north-east of the Netherlands, for the period 1981-2010,
only 30 to 45 days have more than 80 % of sunny conditions[1].
A diachronic analysis of the statistical results revealed two main weather types according to the period. At the
beginning of the 17th century, painters like Avercamp, van Goyen, or van Ruysdael preferred to paint stable and
cold situations. The stratus clouds were dominant. In our corpus of 49 paintings, 17 paintings until 1650 show
stratus (among 27), but only 1 after 1650 (among 22). The water, whether a river, a lake, a marsh, etc., was frozen.
After 1650, cold weather was still dominant: the ice covered all the waters. Yet, the clouds indicated that a weather
front was moving with more cumulonimbus or nimbocumulus. It is also coherent with the amount of people
outside, and with their activities taking place in the winter landscapes. Until ca. 1650, the ice was covered by a
melting-pot of people. Even when the weather was not sunny (with stratus indeed), the people went outside to
enjoy the ice. Throughout the second period, we see less people on the ice and the percentage covered by ice in
the paintings has also changed: 45% until 1650, 35% after. As the painters could imagine, the moving front was
probably passing the land, with windy and less comfortable conditions for people (for example here in Figure 2).

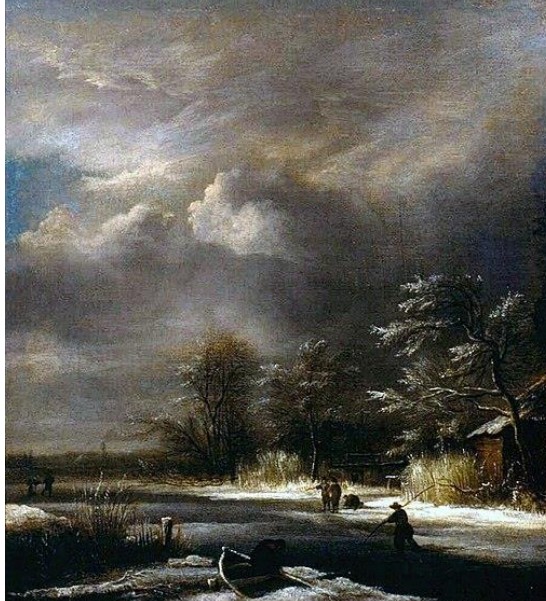


Figure 2: Jacob van Ruisdael, Winter Landscape with skaters, Museum Boijmans van Beuningen, oil on canvas,
37,7 x 33,5 cm.
https://commons.wikimedia.org/wiki/File:Jacob_van_Ruisdael_-_Winter_Landscape_with_Skaters.jpg

Such a difference between the two periods does not reflect the climatic archives, and does not cohere with climatic
variabilities. Two imaginaries of winter were preferred. Indeed, Buisman and van Engelen (Buisman and Engelen,

---

[1] http://www.klimaatatlas.nl/klimaatatlas.php?wel=zon&ws=kaart&wom=Aantal%20dagen%20zeer%20zonnig.



2000) have set up a classification of the cold in winter, from the mildest (1) to the severest (9). In the 17th century,
16 winters were mild, 13 winters were in the average (index 5), and all the other 43 winters were at least "cold"
(index 6, 7 and 8, with no extremely cold winter during our period). Yet, there is no major distinction between the
first, and the second part of the 17th century as Figure 3 shows.

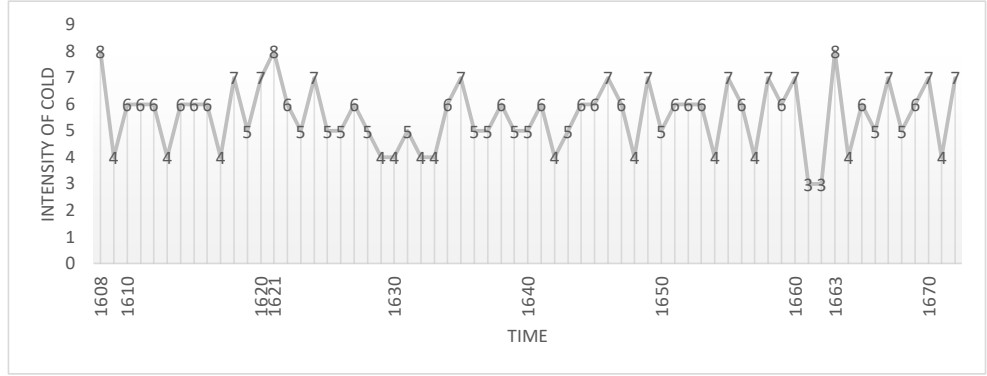


Figure 3: The variability of cold during winter in the Netherlands (1608-1672). 1608, 1621 and 1663 are the most
severe during this period.

This means the frost was not permanent throughout the three winter months (December, January, and February).
For some winters, such as in 1617 and 1670, we have many reports of mild temperatures, trees with flowers in
February, and the impossibility to skate.
Some specific links between climate history and winter landscapes seem to be relevant. On the one hand, 1608
was an extremely cold winter. Several narrative sources mention frozen rivers, such as the Seine, the Tames, the
Rhine, and even the Rhône. In the Netherlands, Dirk Velius, a doctor in Hoorn, reported frost from January 1st
until March. As already emphasized, "Coincidence or not, in 1608, Hendrick Avercamp painted his first dated
winter scene after the extremely hard winter of 1607-08. It would seem unarguable that severe winters like those
of 1664-65 and 1607-08 made an enormous impression on the people and have found a place in the Dutch
collective memory" (Roelofs, 2009, p.29).
On the other hand, there is a scarcity of winter landscapes from the end of the 1620's to the end of the 1630's. This
period also shows relatively mild winters. During the 1630 decade, only 1634, 1635, and 1638 were "cold" winters,
according to Buisman and van Engelen (see table 1). Shortly after this period, both cold winters and winter
landscapes reappeared (Straaten, 1977, p.113). For instance, Jan van Goyen came back to painting winter scenes
in 1641, whereas his last winter scene dated from 1627. The year 1641 also saw the first, dated winter painting by
Aert van der Neer, who then painted ca. 150 canvasses. Curiously not famous in his own time, he was tenant of an
inn, and miserably died in 1677.
Yet, is such a dominance of cold weather coherent with the winter archives?

**3.2.     An imagery of cold weather**





We use the weather compilation made by Jan Buisman as basis for our results. Fabricius' observations from 1594
to 1612 provide us with information about the percentage of "winter" days (Metzger & Tabeaud, 2017). Buisman
used the original source and often provides enough information from Fabricius' notebook to draw pertinent
conclusions as regards weather types. However, Buisman's work is incomplete and some of his approximations
are too imprecise to show how each winter unfolded. For example: "there follows a milder and darker weather
along with rain [January 2nd – January 30th 1596]". Sometimes we must therefore "make do" and suggest
reconstructions despite such approximations taken from a second-hand source reference. To summarise the
sequence of winter weather types, we must rely upon the works of Olivier Cantat (2015). Our reconstruction is
therefore based on a combination of 6 winter weather types taking into account the mention of frost (or not) and
of precipitation (or not).
The following table sums up our results (table 1). We would add that Buisman's written notes did not allow us to
reconstruct these weather types for 10 days in 1598, 11 days in 1599, 6 days in 1660, 6 days in 1601, 7 days in
1603, 2 days in 1604 and 4 days in 1607. It is also needs to be remembered that these uncertainties are sometimes
significant, due to Buisman's approximations. Moreover, unfortunately, Fabricius is the only winter enthusiast
allowing us to make such a kind of reconstruction. Nevertheless, this period being one of the coldest during the
LIA, we can assume the percentage of days without frost was still important throughout the rest of the 17th century.

Table 1: Percentages for the different winter weathers from 1594 to 1612, in December, January, and February
(Buisman and Engelen, 2000)

| A Frost and precipitations | 5,5 % |
|---|---|
| B Frost without precipitations | 16,8 % |
| C Frost-thaw alternation with precipitations | 12 % |
| D Frost-thaw alternation without precipitations | 14,2 % |
| E No frost and precipitations | 26,1 % |
| F No frost without precipitations | 25,4 % |


Several comments can be made. First of all, the most frequent weather types are E and F: on average over half of
winter days had no frost. On the other hand, weather type A is less frequent, which makes sense: cold spells
arriving from Western Europe are brought in by high, even Nordic, continental pressure, causing little to no rain
at all. Next, we notice that the sum of weather types A+B is 22.3%. Lastly, it appears that days with precipitation
(of all types) account for 43.6% (A+C+E). Admittedly, it is possible that Fabricius did not record any precipitation
when there was very little water or snow falling during a day or a night, perhaps because he did not notice such
short showers.
Comparing winter archives and winter paintings showed the painters represented a single kind of winter weather
– frost without rain –, which appeared less than 1/5 days, according to Fabricius' weather records. Furthermore,
they only represented snow in some parts of the paintings, whereas a lot of snow fell during some winter days.
Whenever the cold conditions came after the fall, the snow could remain for a long time. For instance, the weather
enthusiasts spoke about 40 cm of snow in Den Haag in January 1667, and 20 cm in Utrecht in December 1672.





Fabricius also spoke about snow tempests in February 1610, and January 1612. These images of winter have
completely disappeared in the paintings.

### 3.3. A landscape climatic figure? 4 hypotheses

The painters selected a type of winter weather to build an image, even an imaginary of the winter season. Our
image of the LIA is partly transmitted by these paintings, and it is very misleading. The paintings we see in the
museums, on the Web, or in exhibition catalogues, only show specific moments during winters. Why did the
painters represent this image of winter? We now emphasize four different hypotheses.

1) ***Climatic***. The winters were particularly cold at that time. It probably influenced the painters who
   experienced such moments of cold weather. There are some meaningful dates: the 1608 winter was very
   severe, and saw Avercamp's first winter landscape painting. The master of the ice scene was probably
   haunted by the weather he experienced in Kampen, east of Amsterdam. The cold conditions would also
   freeze water more frequently. Curiously, the Zuiderzee is always frozen in Avercamp's paintings. It is an
   image the painter wanted to transmit, even though the see did not freeze every winter, according to the
   written documents. Moreover, as the cold winters were dominant, it meant the winter times were a period
   when water was not an enemy. Frozen water could be tamed and appropriated, whereas liquid water was
   a potential danger because of the impressive floods - even though the Dutch people energetically strove
   to prevent the flooding of their land. More in details: "Only when frozen is water harmless. To celebrate
   this temporary power over the natural enemy, I think it is such an irresistible attraction to the inhabitants
   of our delta. On the ice, the world is literally and figuratively upside down: the catharsis of the national
   primeval goose" (Koolhaas, 2010, p.157).

2) ***Religious***. The Reformation could partly explain this picture of cold winter landscapes. In certain Catholic
   narratives, a severe cold is the devil's place. In Dante's *Divine Comedy*, the frozen places are among the
   most terrible. Moreover, before Brueghel the older's 1565 winter paintings, the few European paintings
   to represent ice used to link the ice to the underworld (such as in Jérôme Bosch's *Garden of Earthly
   Delights*). In the reformed religion, amusements were admitted. It means skating was not seen as a
   depraved activity anymore: Avercamp's pictures "are also the first to include fun on the ice" (Waterman
   Gallery (Amsterdam, Netherlands) and Blankert, 1982, p.22). Furthermore, the landscape genre could
   also be a reaction to the Catholic paintings, more specialized in still lives and portraits: it was a "relatively
   new genre which had no ties with traditional Catholic belief and iconography" (Falkenburg and Finney,
   1999, p.359).

3) ***Political***. The Peace Truce with Spain also played a role. During the war period, as frozen waters did not
   stand as "natural" limits anymore, they could be a disadvantage for the Dutch. A good defensive strategy
   consisted in the artificial flooding of some parts of the territory. However, it was not possible in cold
   conditions, when the water was frozen. Thus, as the Truce between the United Provinces (the
   Netherlands) and Spain was concluded during the very severe month of February 1608, the cold winter
   paintings could be interpreted as a symbol of the celebrated "peace". Frozen waters were not a danger
   anymore. These paintings could also evoke the supremacy of the Dutch, skating in times of war
   (Koolhaas, 2010). Some winter episodes are very famous in the history of the United Provinces, such as



the siege of Haarlem, or Lambert Melisz's 1574 adventure, passing the enemy lines with a skate and a
sledge, together with his mother. Finally, to celebrate winter in the paintings could also contribute to
creating a landscape climatic figure, in contrast with the Spanish landscapes. Indeed, though the Spanish
painters produced less landscape paintings than the Dutch, they represented in majority summer
conditions (Alonso Cano, Juan Bautista Maino, El Greco, Francisco Collantes, Velázquez, etc.) (Gallego,
1968; Pérez Sánchez and Royal Academy of Arts, 1976). To paint so many winter scenes throughout the
Golden Age was possibly, for the Dutch, a way to build their nation after the Spanish domination
(Metzger and Tabeaud, 2015).
4) *Social*. To represent the ice was a way to create and anchor a Dutch identity. On the ice, all the social
classes skated together. The ice was the place where both the rich and the poor walked, played, and so on
(Roelofs, 2009). Furthermore, during the cold winters, it was easier to move in the country. The density
of rivers, lakes, and new canals only enabled the Dutch people to visit their family with skates. For
instance, in 1658, Johan Huydecoper's son visited his family around Amsterdam with his skates (Buisman
and Engelen, 2000): Amstelveen (January 14th-15th), Spaarndam (January 16th), Halfweg and Meresteijn
(January 21st), Bijlmer (24th), Amsterdam (25th), Amstel (30th), Muiden and Naarden (31st), Uithoorn and
Maarseveen (February 1st). One of the written records was held by Caescooper: on December 19th
1676, he skated almost 200 km in 17 hours (Buisman and Engelen, 2006). It is a prefiguration of the
famous Elfstedentocht, in Friesland{Citation}. Without these cold conditions, the Dutch needed a barge
of their own, or to pay for barges and diligences, in order to travel in the Netherlands.
Thus, in François Walter's inventories (2004), icy landscapes could stand as one of the *climatic* landscape figures
to exist in Europe. Indeed, landscapes can represent the nation, and give both political and cultural interpretations
of an identity (Mitchell, 2009).

**4. Discussion**

"The history of climate and culture may be understood as a developing series of conjuncture relationships of great
complexity" (Fischer, 1980, p.246). Thus, we need to be very careful when analysing both fine arts and written
documents. Neither images, nor documents are a true depiction of reality. Painters choose what they would like to
paint, according to both aesthetic criteria, and to the market of Art. In the same way, weather enthusiasts only give
little information about the weather and climatic elements. This explains why the written sources are full of
uncertainties. We have used Fabricius' notebook - described in Buisman's book - to categorize different kinds of
winter weather. Yet, it is very difficult to estimate what Fabricius meant by a "frost day". We have to make with
the available written texts, and can nonetheless suppose the visible marker of frost was the solidification of water
into ice.
By surveying a corpus of paintings and documents, our quantitative methodology tried to overpass such an absence
of exhaustivity. Of course, each corpus - and ours is small - is a scientific construction, and could have been very
different. Moreover, we have considered the paintings as "realistic" depictions of the landscapes. To do so was
necessary to examine, in more details, the climatic elements. Yet, it doesn't mean some messages and symbols in
the paintings are not present or important, as some researchers have argued (Bruyn, 1995). On the contrary, we
agree with the art historians aptly willing to open the interpretations (Buijsen et al., 1993; Sluijter, 1991). The



climatic views on these art masterpieces follow such a recommendation, which enables to discuss the degree of
(climatic) reality provided by the paintings. Thus, "It appears that pictorial content (…) is not something which is
inherently fixed in the image but consists of a « field » of semantic potential which is « triggered » by the image
as well as by the expectations and experiences of the audience" (Falkenburg and Finney, 1999, p.353). If a great
number of winter landscapes were painted, it is also because the Dutch wanted to acquire the paintings, and place
them into their interiors. The public wanted to keep such an imagery at sight.
If only the paintings had been analysed, our knowledge of past winters during the LIA would be too specific, and
thus, incorrect: "In some respects, contemporary Dutch winter landscapes are a uniquely problematic source for
cultural historians of climate change, despite their seemingly obvious connection to the Little Ice Age" (Degroot,
2014, p.457). Therefore, we disagree with the statement according to which "even if they are not necessarily
indicatives of cold climate, the effects of these winters are well represented in historical paintings, e.g. in « Winter
landscape with skaters », by Hendrick Avercamp" (Diodato and Bellocchi, 2012). The written documents also
mention cases of people who died of the cold, passed through the ice, and cases of plague during the Golden Age
(Buisman and Engelen, 2000). In some proverbs, the ice is depicted as something both enjoyable and dangerous
(Straaten, 1977). At least in Avercamp and some of his contemporaries' paintings, there is no such an image of
winter bringing death (cf. fig. 4). However, some other paintings represent a less enjoyable image of winter. Hence,
it would be very interesting to understand why it is Avercamp's winter landscapes "which, four hundred years on,
still govern our image of the cold winters of the seventeenth century" (Roelofs, 2009, p.83).

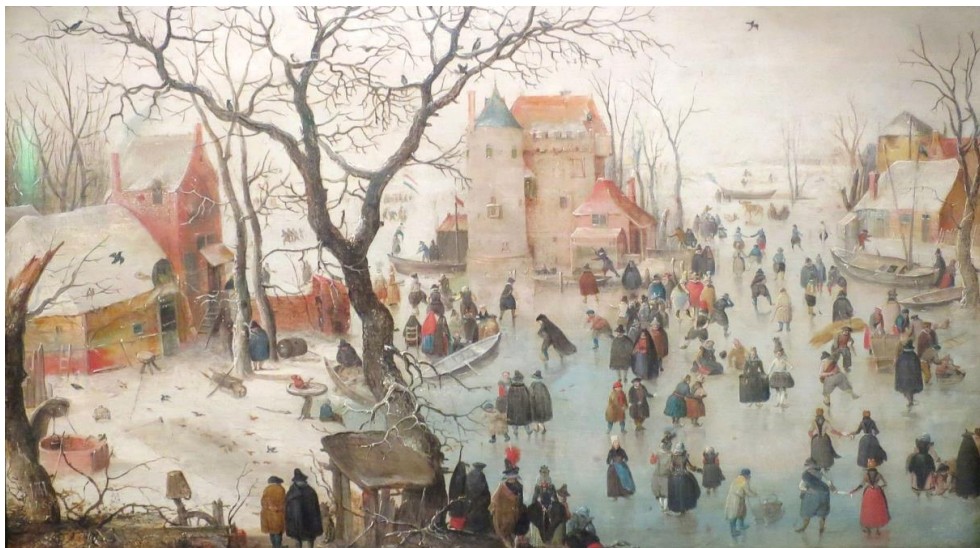


Figure 4: Hendrick Avercamp, Winter Landscape with Ice skaters, 1608, Art Museums of Bergen, oil.
https://commons.wikimedia.org/wiki/File:Winter_Landscape_with_Ice_Skaters_by_Hendrick_Avercamp,_Berg
en_Kunstmuseum.JPG

Is it because nowadays, we are used to seeing people enjoying the ice? Or, is it because these paintings remain
national images of the Netherlands? Like the majority of Dutch painters, Avercamp represented the ice, and
skating. Undoubtedly, skating remains today an essential component of the Dutch identity. The Dutch skaters often
reach the top position in the Olympic games, whereas despite climate change, the Elfstedentocht (which last took
place in 1997) is still an emblematic challenge.
The winter landscapes fashion probably resulted from a combination of climatic, political, social, and religious
facts. The four hypotheses we have discussed are imbricated. However, the winter landscape genre also exists
because of other factors. After a blooming period of winter scenes, the painters decided to change their themes
(Robinson, 2005). Moreover, shortly after the 1670 decade, the Golden Age ended up (Denys and Paresys, 2016;
Prak, 2009), the market of art declined, and many painters died – e.g. Jacob van Ruisdael in 1682. Though the end
of the 17th century was one of the coldest phases during the LIA, with the Maunder minimum (Diodato and
Bellocchi, 2012; Wanner et al., 1995), winter landscape paintings were neglected.

**5.  Conclusion**

Winter landscapes during the Dutch Golden Age represent certain kinds of weather. They do not make a reliable
source for climatic data. Yet, they stand as testimonies of severe winters, and show the societies' adaptation to
certain cold conditions. Moreover, they inform us about the cultural depictions -and related choices- of weather,
and by extension, of climate. The painters have created a "picture" of winter. The paintings are not only images,
as their authors have created a highly partial and sketchy imagery of -a period during- the LIA. Thus, to better
understand the climatic images and imageries of societies, interdisciplinary approaches between the history of art,
the history of climate, and climatology, need to be extended. This remains an area that requires further
investigation.

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
