# Peer review of "Enjoying the ice. Dutch Winter landscapes, weather and climate in the Golden Age, 17th century"

_Climate of the Past, 2020_

## Referee Comment (RC1) · Anonymous Referee #1 · 10 Jul 2020

The "winter landscapes" crafted by sixteenth- and seventeenth-century artists in the Low Countries – especially the Dutch Republic – have become the quintessential cultural expressions of the Little Ice Age. For decades, scholars in a range of disciplines have considered what they reveal about weather and climate in the period – and what they suggest about human responses to shifting environmental conditions.

This article, then, should be considered in the context of a large corpus of scholarship. The question is: does it say anything truly new? The answer: it might, if greatly refined. The paper does not have a coherent argument, but rather asks a question: do seventeenth-century winter landscapes actually reveal winter – and thus the Little Ice Age – as it was? This is not an original question, nor is the author's means of answering it – to compare weather reports in written sources with weather depictions in paintings

[Figure]

– but the answer is pursued somewhat more systematically in this article than it has been in other studies. In particular, the statistical approach used in this article strikes me – as a non-statistician – as intriguing, though it is based on a small sample (more on that below).

Indeed the paper has several serious shortcomings, which I will identify by order of severity, from most to least serious.

Most importantly, its method is clearly problematic. The author analyzes a sample of 49 winter landscapes, but the only reason for choosing that sample seems to be that they were in an international exhibition. The reason for not selecting more paintings seems to be that some have been lost or are difficult to access. While it is not necessary for the author to examine more paintings, a more convincing explanation should be given for the size of the sample.

The author attempts to compare winter weather in this selection of paintings to climate and weather as they really were. Because reconstructions that use "archives natural" do not reveal seasonal or monthly weather, he has decided to use weather diaries written by weather observers, especially David Fabricius. Yet the author has not himself examined these primary sources, but rather relied on descriptions of those sources by Jan Buisman, the author of several volumes that summarize Dutch weather observations.

There are several related problems with this approach. First, it is just not true that paleoclimatic proxy reconstructions and (not considered by the author) model hindcasts can shed no light at all on cold conditions in the seventeenth-century Low Countries. The only paleoclimatologist cited in this article is, I believe, Ulf Büntgen, and the only publication cited is an overview of the Little Ice Age written for a history journal. That is insufficient. In my view, the paleoclimatic record could have enriched the author's description of winter weather and climate in the seventeenth-century Low Countries. At the very least, the author should cite paleoclimatologists – not just historians – to

explain why he decided not to use that record.

Second, the author himself admits that consulting Buisman – rather than the primary sources Buisman summarizes – limits his analysis. The author writes that "sometimes we must therefore 'make do' and suggest reconstructions despite [Buisman's imprecise] approximations." That is not acceptable – not only because it is unnecessary (the author could have consulted the primary sources), but also because it potentially invalidates the statistics in table 1 (the statistics that ostensibly allow the author to discern whether painters depicted real weather). The author admits that "uncertainties" in these statistics "are sometimes significant, due to Buisman's approximations." In fact, it is due to the author's decision not to consult the primary sources Buisman described.

Third, it is generally advisable for scholars to avoid using weather diaries by themselves to reconstruct past trends in weather and climate. As the author (to his credit) acknowledges, a diarist such as Fabricius did not necessarily record every change in weather, and weather diaries are inevitably biased in favor of spectacular weather. A diarist might note a storm, but not a sprinkling of rain. Over time, such omissions can skew attempts at climate reconstruction. Nor should scholars assume that weather that prevailed in one part of even a small country, like the seventeenth-century Dutch Republic, necessarily affected another part. The reason that painters did not represent a weather event recorded in a diary may simply be that that weather event did not affect their locality. The way around these problems would have been for the author to consult more sources, including not only direct written reports of weather on land but also descriptions of activities that must have been affected by weather, or observations of weather at sea, or the paleoclimatic record, for example.

The lack of references in this article to paleoclimatic studies is indicative of a larger failure to engage with the scholarship not only of climate history, but also specifically of the climate history of Dutch winter landscapes. Most seriously, the author does not cite Dagomar Degroot, The Frigid Golden Age: Climate Change, the Little Ice Age, and the Dutch Republic, 1560-1720 (Cambridge University Press, 2018), although he

does cite Degroot's dissertation. The book contains extensive passages that explore whether Dutch winter landscapes really depicted weather as it was. While this article has the potential to build on those passages, there are parts of the article that present a less detailed and less complete overview of issues already considered in The Frigid Golden Age. That book also references scholarship on winter landscapes that should be – but is presently not – cited in this article.

More broadly, the author supports claims about the Little Ice Age, the character of climate history, and the link between climatic and social histories using a small selection of publications, all at least seven years old. Given the explosion of new scholarship in climate history, far more publications should be cited, by more diverse authors. Claims about the nature of the LIA should moreover reference paleoclimatologists and historical climatologists, not historians citing scholars in these disciplines.

The article also suffers from a host of minor factual errors and omissions. For example: on page 1, the Dutch did not just experience the LIA during the seventeenth century (but for them that century was likely the coldest of the LIA), and Brueghel the Elder (not the Older) is probably the best-known painter of winter landscapes; on page 2, the Dutch Golden Age (itself an increasingly loaded term) did not necessarily "come to an end" with the 1672 rampjaar (it is a defensible argument but not one universally accepted by historians); on page 3, the "scientific knowledge" we have of the seventeenth century is not spelled out, and a reference to present-day Dutch climate is irrelevant given how much that climate has changed; on page 4, it is not clear why the painting must depict an afternoon scene because the artist has chosen to depict many people; on page 5, it is not clear why we should care that people today care about weather (as opposed to people in the early modern past, whose attitudes scholars have explored); on page 5, the significance of a 10% shift in the "percentage covered by ice in the paintings" (itself unclear) is hard to gauge given the small sample; on page 5-6, the means of classifying winter cold should be clearly explained; on page 6, the graph does not represent the two "parts" of the seventeenth century, and the caption should

cite the source (it also requires better labels); on page 8, it is simply inaccurate that frozen water posed less of a threat to the Dutch than liquid water, given for example the peril of invasions over frozen waterways (as the author acknowledges), the risks imposed by frozen waterways, and the threat of ice dams (all clearly described in The Frigid Golden Age); on the same page, descriptions of religious and political impacts seem very simplistic; on page 9, it is simply not true that it was easier to move through the Republic in "cold winters" than in other conditions, considering the centrality of waterborne transport via canals, rivers, and the Zuiderzee. In general, the "hypotheses" present by the author for the weather conditions omitted by painters from their winter landscapes consist of vague speculation that could be enriched with deeper primary source research.

Finally, a number of typos and mistranslated expressions undermine the article. There are also a couple non sequiturs (when we find out, for example, that Aert van der Neer never achieved fame and "miserably died" in 1677).

To conclude, this article is based on an interesting idea, and the method it proposes could yield interesting results. It is fascinating to consider which weather conditions seventeenth-century artists chose to depict, and which they did not. Yet for this piece to deserve publication in Climate of the Past, it needs a stronger argument, a more convincing method, a deeper engagement with existing scholarship, a stronger grounding in primary source research, and a series of minor factual corrections. I believe the author can do it, but it will take substantial work.

---

## Referee Comment (RC2) · Anonymous Referee #2 · 11 Oct 2020

1) General comments

This is a very interesting contribution to the use of landscape paintings as possible source of the reconstruction of winter weather in the 17th century. The author deals with an important issue in general, because Dutch winter paintings of the 17th century are often used to illustrate publications on Little Ice Age without further source criticism. It is important to contextualize the paintings in their climatological cultural background and the author succeeds at least to some extent in doing so by analysing a corpus of 49 paintings from the 17th century Netherlands. Based on documentary evidence (the collection by Buisman and van Engelen as well as studying the weather observations by "enthusiasts" such as David Fabricius) he carves out that the weather situation depicted (cold winters with frozen rivers, canals, lakes and the sea, but no

precipitation) is only one pattern out of six. According to Fabricius's observations, this weather pattern only represents about a sixth of all winter days. At least some of the winter landscapes refer to really cold winters with freezing canals, lakes and sea (e.g. in 1608, when Hendrick Avercamp painted his first winter landscape). The popularity of winter landscapes corresponds with the cold period of the Maunder Minimum, but it represents also political and social circumstances of that time. However, as will be shown in the following section, the arguments used by the author are not always convincing. Furthermore, several errors have to be corrected before publication. Some publications both on the overall climatic development of the Little Ice Age as well as on the use of paintings for historical climatology should be recognized.

2) Specific comments on individual scientific issues

Chapter 1: Introduction l. 22: "medieval warming" is a concept that has been rejected more and more in recent research – see, for instance, the Palgrave Handbook of Climate History (2018,the new authoritative handbook on historical climatology that has not been cited at all). The whole paragraph has to be elaborated in more detail, also pointing out that Little Ice Age has not been a period with only cool weather conditions but significant fluctuations. In the context of this article, the Maunder Minimum should be mentioned and explained here for the first time and not only very briefly at the very end on p. 11 (l. 336). l. 45-47 The remark that only a few studies have explored European landscape paintings with climatic analyses is only partly true. There is, indeed, quite a potential for more studies, but there are several studies the author neglected, e.g.: Brönnimann, S.: Picturing climate change. In: Climate Research 22 (2002): 87-95. Thornes, J.E.: Constable and the Beaufort Wind Scale. In: Hamilton, J. (Ed.), Fields of Influence. Birmingham 2001: 93-109 Thornes, J.E.: Cultural climatology and the representation of sky, atmosphere, weather and climate in selected art works of Constable, Monet and Eliasson. In: Geoforum 39 (2008): 570–580. Rohr, C.: Die Winterbilder der französischen Impressionisten im Spiegel der Klimageschichte. In: Kornhoff, O. (Ed.): Lichtgestöber. Der Winter im Impressionismus. Arp Museum

Bahnhof Rolandseck, 11. November 2012 bis 14. April 2013. Bielefeld 2012: 48-54.

Chapter 2.1 A corpus of winter landscapes l. 84 The wording "cloud classification was only invented in 1802" obviously refers to Luka Lamark's system. However, single weather observers sometimes developed their own system to distinguish between different types of clouds. I suggest rather speaking of a "commonly used and standardized cloud classification" l. 89: The caption of Fig. 1 contains some errors: read "Salomon van Ruysdael" instead of "Salomon van Rusydael" and "New York, Metropolitan Museum of Art" instead of "Metropolitain Museum of Art" without naming the place of the museum. l. 93: Here, Salomon van Ruysdael and Jacob van Ruisdael cited later are mixed up. Correct is "Salomon van Ruysdael" instead of "Jacob van Ruysdael" l. 94-95: I cannot follow the argument that people are moving toward the inside "Which means the tide is reasonable". If the water is completely frozen, the tide will not cause the people moving. In addition, we absolutely do not know whether van Ruysdael depicted the sea (e.g., the Zuiderzee) or a larger frozen water area inland or a larger canal or river. The painting is named "Drawing the Eel" and obviously refers to the eel pulling festival (the so-called palingtrekken). Maybe the author could think about a more specific localisation based on the tradition of palingtrekken.

Chapter 2.2 Climatic sources The collection of sources by Buisman and van Engelen is, as the author states himself, a good starting point, but not always satisfying. Including David Fabricius in more detail is without doubt a good choice, but still it would need more information about freezing canals etc. I wonder why the author did not refer to the accurate study by Adriaan de Kraker of 2016 (Ice and water. The removal of ice on waterways in the Low Countries, 1330–1800. In: Water History) dealing with the ice removal on Dutch and Belgian canals (amongst others the Leiden – Haarlem – Amsterdam canal). This study shows that in particular the canals got frozen during Little Ice Age quite frequently, but that people tried to break up the ice due to several reasons.

Chapter 3.1 Painting and climatic diachronies l. 152 Please add the place of the Museum Boijmans van Beuningen (i.e. "Rotterdam") in the caption of Fig. 2. l. 172 read "Thames" instead of "Tames" l. 180: The reference "see table 1" is obviously wrong. It should rather be "see Fig. 3".

Chapter 3.2 An imaginary of cold weather l. 193-194: this is obviously a quote taken from Buisman, but the reference is missing. l. 200: read "6 days in 1600" instead of "6 days in 1660" l. 217-223: The last paragraph of the chapter is a bit confusing. First, it starts with a reference to Fabricius. Second, an information on 1667 and 1672 is inserted. Finally, the narrative turns back to Fabricius. Please rephrase in a logical order.

Chapter 3.3 A landscape climatic figure? 4 hypotheses l. 225: The title "A landscape climatic figure?" is not understandable to me. Please revise wording. l. 228: I cannot see, why the image of winter given by the paintings is "very misleading". It is indeed "one-sided", but we have to ask, why painters should have produced also landscapes with rainy winter weather or other non-spectacular winter scenery. This is a question of aesthetics and of the market of art. l. 244-253: I totally disagree with the religious argument and I suggest skipping it totally. Having in mind that the reformed churches in the northern Netherlands were mostly Calvinists, this is the church fighting against amusements like ample eating and drinking or dancing more than any other confession. Catholics, however, were allowed to celebrate with only few restrictions, if there was a celebration day, a public feast etc. The examples given for a negative winter experience seem to be taken by pure chance and we have plenty of opposite examples both for the Middle Ages and for Early Modern Times (e.g. frescos of people enjoying snow in winter by making snowball fights, as in the cycle of months in the Torre dell'Aquila in Trento, Italy, around 1410). Without doubt, winter had also been a symbol for the old age and even death, but not related to a specific confession. And people who were freezing would not have enjoyed winter at all, no matter if they were Catholics, Lutherans, Calvinists or Non-Christians. l. 255-256: For artificial flooding as a defensive strategy in the Netherlands, one should refer to Adriaan de Kraker's article "Flooding

in river mouths: human caused or natural events? Five centuries of flooding events in the SW Netherlands, 1500-2000" (Hydrology and Earth System Sciences 19/6 (2015): 2673-2684, in particular 2678 and 2680-2682). l. 279: A Citation is missing

Chapter 4: Discussion l. 290: Only here, the aspect of the market of art and its demands is mentioned shortly. This aspect should be discussed in more detail, because it explains best why so many winter landscapes had been painted (e.g. by Hendrick Avercamp). l. 291: I would rather speak of "limited information" instead of "little information" given by weather enthusiasts, because compared to other sources for climate reconstruction of that time they are actually quite rich sources.

3) Technical corrections

a) Language and style

Both language and style are satisfying. Nevertheless, the paper should once again be proofread by a professional native speaker. For instance, the term "precipitation" should only used in singular (instead of "precipitations" as used six times in Table 1 ,l. 208). Similarly, the term "weather" should always be used without article (with article in l. 76, 82, 113, 119, 124, 146, 234, 291). Some typos should be corrected, such as "see" instead of correctly "sea" (l. 236). Please check the passages quoted verbatim once again, e.g. read "natural archives" instead of "archives natural" (l. 109). In addition, I suggest avoiding "we" for a single-authored paper, if only the position of the author is expressed (l. 12, 38, 52, 53, 57, 60, 61, 71, 73, 76, 82, 104, 125, 134, 147, 189, 199, 230, 292, 298, 300, 311, 332. In the same sense, avoid "our"/"ours" (l. 41, 54, 71, 77, 109, 134, 141, 160, 189, 196, 199, 227, 242, 296, 297).

" b) Formal requirements

l. 26 and 417: F. Mauelshagen's "Klimageschichte der Neuzeit" appeared in 2010 and not 2012. The place of publication is "Darmstadt" and not "s.l."

Some smaller corrections to be made: Read "Janković" instead of "Jankovic" (l. 42,

406) "van" is a fixed part of Dutch surnames. So, always read "van Engelen" instead of "Engelen" (l. 126, 157, 207, 275, 278, 315, 367, 369)

4) Overall assessment

In general, this is an interesting contribution addressing an important question for historical climatology, which should be accepted with major revision, i.e. there are both some critical remarks in content to be dealt with and in addition, some technical improvements (formal requirements, typos) and some clarifications needed, as mentioned in my comments.

---

## Referee Comment (RC3) · Anonymous Referee #3 · 20 Oct 2020

This article evaluates the relationship between climate, winter weather, and its potential connections to the Dutch winter landscape genre. It argues that seventeenth-century winter landscapes overrepresented cold, dry weather based on an analysis of contemporary weather narratives. It tentatively explores several non-climatic reasons why artists might have chosen to emphasize these icy scenes. This article is intriguing, in part, because it promises to contribute to a vein of historical, climatological, and art historical scholarship dating back to at least the 1960s. As it currently stands, this article only superficially engages that literature. It suffers two primary limitations as a result. It lacks a clear explanation of its broader relevance and relies on source material with a less than critical acknowledgement of its potential and limitations. To be clear, this is a promising article that warrants further investigation. If accepted, I recommend the

author consider the following.

Broader relevance: The author presents their study as a semi-quantitative evaluation of artistic decision-making. Why did artists choose to depict the scenes they did? Was it motivated by their experience of weather or some combination of other factors? In the aggregate, can we detect subtle shifts in climate by evaluating the prevalence of environmental conditions in winter landscape corpora over time? The author is not the first scholar to address these questions, and to their credit they reference several important studies, beginning with Burroughs in 1981 and ending with Degroot's dissertation (though strangely not his book). Hans Neuberger was addressing these ideas in 1970 and the climatologist H.H. Lamb proposed using landscapes to evaluate the prevailing character of changing summers and winters in 1967 and 1977. Bonacina was already exploring similar issues in 1939. These early efforts were not unproblematic and subsequent literature (some of which is cited) refined their methods and identified important limitations. While I don't advocate the author cite all of these sources, it may be useful to think about how this research program has change over time and where, specifically, the author sees their own contributions intersecting its most important concerns. In more recent years, the consideration of winter landscape paintings as source material for climate history has periodically reemerged, whether in the Journal of Interdisciplinary History (Kelly and Ó Gráda, 2014 and its critique by White, 2014) or Dagomar Degroot's Frigid Golden Age (2018). Scholars have been interested in the relationship between winter landscapes production and climate/weather for more than fifty years. The longevity of this project speaks to its ongoing relevance, but it can't be left to speak for itself. The author needs to clearly explain their intervention(s). Is this simply an attempt to assess the usefulness of landscape paintings as source material for historical climatological reconstructions? If so, what does it add to previous findings? Is it making new claims that challenge the notion that Dutch landscape artists reliably documented or 'described' environmental conditions?

The author claims that this last question has only been "partially encompassed" (56),

but Dutch landscapes' representational fidelity has been a subject of interest (and sometimes debate) since at least the 1980s. Wayne Franits Looking at seventeenth-century Dutch art: realism reconsidered (1997) is a good, though somewhat dated introduction. My reading of this literature is that few art historians (or climate historians for that matter) would argue these paintings reliably recorded weather as it was experienced. This does not diminish their value for studies of climate perception or the influence of climate on culture, though it does not appear the author sees their work contributing to these latter themes. Regardless of whether the author intends their work to add to one or both of these approaches, it needs to be stated more clearly at its outset. A more substantial (and critical) evaluation of this shared historiography would more clearly point to the purpose and significance of the article, which is currently framed as a set of questions (55-56; 185; 229-230) more than an argument.

Source material: This study is based on a corpus of 49 winter landscape paintings. While I understand that an exhaustive accounting is impossible, is the Suchtelen exhibition catalog the most complete available? The answer may simply be "yes", though it's unclear if this is the case. I'm also a little curious why the author chose to restrict their study to painting. Winter landscapes appeared as prints as well, for instance. Is this because prints are not detailed or atmospheric enough to project "weathered reality?" I don't necessarily think this study should include prints, but a richer justification of the value of painting as a medium for this type of analysis might be warranted. The reliance on written narratives (mostly compiled by Buisman) to reconstruct "actual" climate is at once rich with potential and frustratingly underdeveloped. Written narratives are certainly not the only sources available to us to reconstruct historical climates. The author might have turned to historical climatological reconstructions in combination with these chronicles, letters, and weather observations either compiled by Buisman or straight from the source.

The approach as it currently stands is rich with potential because so many of these narrative descriptions suffer from similar limitations as the paintings (written often long

after the fact, indoors and presumably insulated from the weather, based on second-hand accounts, subject to the conventions of textual genres). There is a fascinating opportunity here, I think, to critically address the strengths and the weaknesses of both sets of sources when put in conversation with one another. The author indicates as much in the discussion ("Neither images, nor documents are a true depiction of reality" (289)) If taking this approach, the author should use the original source material rather than Buisman's selections removed from context. This might also go a long way to substantiating the hypotheses in section 3.3. If LIA paintings are misleading (or an 'imaginary'), might these textual sources support these claims as well?

The hypotheses themselves are intriguing because they begin this conversation, but they remain underdeveloped. The "Climatic" section relies on one scholar's reading of ice as a "tamed and appropriated" nature. Naturally, water could be a hazard in the Dutch Republic, but it was equally an ally. The same could be said of ice and as Degroot demonstrated in his Frigid Golden Age, frozen rivers, lakes, and seas provided benefits and disadvantages. Perhaps a closer evaluation of artists like Avercamp and their social milieu would indicate whether they were likely to view ice as friend or foe.

The "religious" section is the closest we come to an iconographical or iconological reading of ice scenes. Dutch paintings often evoked political, religious, and moral meaning and scholars following Panofsky have used literature like emblem books to examine these symbolic meanings. The author agrees these interpretations have merit and cites Buijsen and Sluijter, yet doesn't interpret these scenes iconographically. The author, in fact, seems to partly discount these readings by arguing that the Reformed Dutch dispelled Catholic associations of ice with the underworld. Landscapes may have lost their ties to Catholic iconography, but did they not reflect Reformed beliefs? Skating, at least according to Van Suchetelen (2001), symbolized recklessness and the transience of life for instance. While I wouldn't advocate a deep reading of every painting, it would certainly enrich these hypotheses to present a greater diversity of interpretations.

I am also left questioning whether there are other potentially significant explanations for these winter weather imaginaries. The most obvious are the art market and the influence of artistic convention. Consumers of paintings expected to see certain stylistic and formal elements included in these paintings. Although it is fascinating to read about the meteorological coherence of Salomon van Ruysdael's (incorrectly attributed to Jacob van Ruisdael) "Drawing the Eel", just as often artists did not adhere to this degree of realism. This was the finding of John Walsh, whose chapter on 'Skies and Reality in Dutch Landscape' in Art in History, History in Art (1991), which is strangely absent. Indeed, Walsh singles out Salomon van Ruysdael for his striking and realistic depictions of clouds as well as their unrealistic placement and appearance in Dutch skies. He concludes these were hybrid atmospherics, partly reflecting observation, partly influenced by other artists, and partly reflecting to the formal requirements of the genre. Dutch artists may have simply deemphasized precipitation and emphasized frost because this was their (or their buyers) aesthetic preference.

The same note of caution to taking these paintings at face value would apply to other formal elements beyond clouds. In other words, the finding that "an element identified in one painting often appears in the other paintings as well" (75) MAY indicate useful information about climate, but also any number of other reasons completely unrelated to even these religious, social, or political hypotheses. I also wonder whether any changes in the art market may have influenced the timing of peaks and nadirs in the production of winter landscapes. Jan de Vries and Ad van der Woude each contributed a chapter to the above volume that may point in productive directions. Finally, I'm curious about the diversity of winter landscapists represented in the corpora. Does Van Suchtelen claim the catalog is representative? (71) The author emphasizes Avercamp and notes work by Van Ruysdael, Van Goyen, and a few others. Aert van der Neer apparently painted 150 canvases, though the majority are obviously not included in this study. I'm curious because if one or several artists are disproportionately represented, then would not their artistic conventions (also the ups and downs of their careers) have outsized influence on the study? Avercamp died in 1634 during (perhaps not

coincidentally?) one of the lulls in winter landscape production. The periodicity of artistic production that the author identifies is among its more intriguing findings, but without a clearer sense of representation in the sample, the reader is left wondering what accounts for these changes.

Minor issues: The characterization of the Little Ice Age as experienced in the Dutch Republic is problematically uniform. There is little indication that temperature and precipitation underwent substantial decadal change during the period they investigate, which of course it did. An oversight considering the relevance of decadal change in painting/climate diachronies. (23-26)

It's a bit of a stretch to so definitively claim that the Golden Age ended in 1672. The art market collapsed that year, perhaps a better justification? (39-40)

I'm not sure "cultural appropriation" is the correct term (42)

The claim that "most geographers looking at paintings either draw on cultural geography, or visual studies also seems to be a stretch – see, for instance, work on the 1674 derecho that struck Utrecht (Gerard van der Schrier and Rob Groenland, "A reconstruction of 1 August 1674 thunderstorms over the Low Countries") or Baart et al "Using 18th century storm-surge data from the Dutch Coast to improve the confidence in flood-risk estimates".

The sentence "For instance, Jan van Goyen came back to painting winter scenes in 1641, whereas his last winter scene dated from 1627" is confusing and could simply be reworded as "Jan van Goyen painted no winter scenes between 1627 and 1641."

There are also numerous grammatical and stylistic issues as well as spelling mistakes (for instance, Table 1: changes from "precipitations" to "precipitation", sea is misspelled (236), the quote that begins on 311 does not have a closing quotation mark)

In summary, there is rich potential in this article. The author is indeed correct that visual representations of climate are rich in interpretive possibility. Their study poses

a number of fascinating interventions in this venerable tradition. With a clearer elaboration of the study's place in that literature and more critical use of textual and visual documentation, both of which will take substantial work, this article would be better positioned to find a place in Climate of the Past.

———————————————————

---

## Author Comment (AC1) · 21 Oct 2020

General comments

I would like to thank the three reviewers for their remarks. All of them invite to a substantial work with major revisions. I try to answer here the main points that are, for R1, R2, R3 or all reviewers problematic.

Specific comments

1)Written and paleoclimatic sources

About the written source first, I would have to use very precise information about the weather, if possible at a daily scale. As my main objective is to show if yes or no the

winter paintings have truly depicted the weather conditions during this phase of the LIA, I think I cannot use other paleoclimatic data with sufficient precisions regarding the time scale. Tree rings may be useful to estimate some general climatic condition each year in the past but certainly not at a daily scale. As Christian Pfister, Pierre Alexandre or more recently Dagomar Degroot have demonstrated, written sources are the only paleoclimatic data showing climatic variabilities during days, weeks... So, I disagree with the comment of R1 "In my view, the paleoclimatic record could have enriched the author's description of winter weather and climate in the seventeenth-century Low Countries. At the very least, the author should cite paleoclimatologists – not just historians". My purpose is to show the main weather conditions that are represented in the paintings and to interrogate this representation, this imagery.

2) Fabricius' diary and Buisman's books

I am completely sure that the original source (Fabricius' diary) would have been precious and more relevant for this paper. Buisman's books are a secondary source and cannot have the precision of the diary. But since my PhD I have tried to find some colleagues or students who would like to analyse Fabricius'diary and unfortunately it was not efficient. In fact, I have scanned all the diary (more than 350 pages) but it is very hard to read it. Even colleagues in medieval studies could not help me and I have no the good eyes to read that kind of source. I think it would be fantastic for a postdoc or a PhD student to decipher and analyse the diary but I don't have a position allowing that engagement. Attached is reproduced one page of the diary. So, it is not "due to the author's decision not to consult the primary sources Buisman described" (R1). So, I have decided to use Buisman's books but I must say that Fabricius' diary is quoted with a high degree of precision. Buisman has not reproduced all the diary but give, I am quite sure of that, sufficient information to produce the little statistic analyse of winter weather I produce in table 1. Even if it is not the same period, de Kraaker showed annual number of ice coverage days of canals in and around Haarlem (1670–1730) – cf. comment of R2. It was also during a cold phase of the LIA. His analyse shows that

for this period canals were covered by ice during ca. 28 days (average of the period). I can quite seriously suppose that during March and even April some days were also frozen. Hence, as I focus only on the 3 Âń winter Âż months (DJF) in my paper, the average would be lower, maybe around 20 days (for 89 days DJF). It is quite similar with the percentage of frost days given by my analysis of Fabricius' diary in Buisman's book (22%).

3) Other sources and Degroot's studies

Of course, to respond R1, I have consulted a lot of different sources and not only Buisman's book. Degroot's Dissertation and book are rich of different sources showing how cold is experiencing by Dutch people, notably during Wars. But it was not in this paper the purpose to compare the sources and give other precisions about the diversity of winter experiences and testimonies. Furthermore, I disagree with R1 when he/she writes that : "Most seriously, the author does not cite Dagomar Degroot, The Frigid Golden Age: Climate Change, the Little Ice Age, and the Dutch Republic, 1560-1720 (Cambridge University Press, 2018), although he does cite Degroot's dissertation. The book contains extensive passages that explore whether Dutch winter landscapes really depicted weather as it was". Of course, I have read this very interesting book but it doesn't show if the weather is truly depicted in the paintings. P. 264-268 for example there is no description of the weather, of the clouds, the snow, the ice. . . Degroot writes that "we can therefore consider them genuine cultural responses to the Little Ice Age" p. 267. In this book, cultural testimonies of the LIA are not looked with a geo-climatologist lens. That is what a would like to add in my demonstration. Moreover, I disagree with the remark "The reason that painters did not represent a weather event recorded in a diary may simply be that that weather event did not affect their locality" (R1). It is not convincing, considering the synoptic climate of the Netherlands. If Fabricius lived in a particular place, he has with any doubt experimented the diversity of winter weather that the painters (in Amsterdam, Haarlem, Kampen. . .) have also experienced. I have consulted a lot of documents written by the KNMI and it is clear that even if they

are some gradients of temperature and precipitation in the Netherlands, the climate in Aurich (where Fabricius lived) and Amsterdam and the surroundings are not very different.

4)The corpus of paintings

Indeed, the corpus of paintings is limited and it is always difficult to build a corpus. I could have selected other paintings and I wondered if it was better to build only a corpus of Avercamps's paintings, on paintings in one Museum, on paintings more specific to a period during the 17th century. . . I have weighed the pros and cons of this choices. But it appears that the exhibition in 2000-2001 showed a certain diversity of artists. I cannot demonstrate it, but I have looked at more than 1000 Dutch winter paintings in different exhibition catalogues, museums. . . I think that they are quite representative but I don't know ho to prove it more scientifically... Avercamp is not overrepresented (4 paintings) and Aert van der Neer is included (also 3 paintings). Prints were not added in the corpus in order to maintain a certain coherence (R3). For sure there is a high amount of winter landscape prints. . . But sometimes the skies (and the clouds) are not represented and the interpretation would be more restrictive. So, I hope to answer the commentary "While it is not necessary for the author to examine more paintings, a more convincing explanation should be given for the size of the sample" and "I'm curious because if one or several artists are disproportionately represented, then would not their artistic conventions (also the ups and downs of their careers) have outsized influence on the study?" (R3). I think it is not the case and I propose to add the list of 49 paintings in a supplementary material.

5) State of the art – climate reconstructions with paintings

I am totally ready to add more information about different previous works that have question Dutch landscapes or more generally paintings and past images for climatic reconstructions (R2). It is one of my research field and I have published papers (in French) focusing on other paintings, related with climate issues (Sisley and the floods,

Alexandre Hogue and the Dustbowl. . .). I also have discussed the different methods useful to analyse landscape paintings with climatic interpretations. It could be added in this paper and I totally agree with R3 when he/she writes that "it may be useful to think about how this research program has change over time and where, specifically, the author sees their own contributions intersecting its most important concerns". But I am not sure that, as R1 writes, "Given the explosion of new scholarship in climate history, far more publications should be cited, by more diverse authors". What is the utility of citing more and more authors instead of focusing in the main works? About J Walsh's analyses (R3), I am not totally convinced by his remarks and commentaries about the reality of skies in Dutch paintings. With closer eyes, I am often quite sure that the skies could be Ân real Âż. Looking at the different clouds, it is totally coherent and van Ruysdael's painting in fig. 1 (sorry for the mistake you are absolutely right) is not an exception. Here, on the contrary, I agree with Stanley David Gedzelman and his analyses of Dutch skies. He shows with very convincing arguments how they could be explained by our contemporary knowledges of weather patterns.

6) The religious argument

About the different arguments in part 4, I think that the religious argument is not removable, even if it has to be more nuanced. I agree with R2 when he/she writes that "The examples given for a negative winter experience seem to be taken by pure chance and we have plenty of opposite examples both for the Middle Ages and for Early Modern Times". I have looked at different books showing how winter was depicted in the medieval images (e.g. Pearsall and Salter, see also my paper here : https://www.projetsdepaysage.fr/premi_res_neiges_le_paysage_d_hiver_dans_les_enluminures_xive_xvie_si_cle_). But I emphasize that numerous images of snow appeared (in books of hours for example) but never icy landscapes. So, the imagery of "enjoyable" winter has probably changed between the medieval and catholic time (snow has positive values in different catholic texts) and the 17th century in the Reformed Netherlands. Now, ice seems to be appropriated, even if symbolic images and interpretations are possible. Calvin

himself wrote that leisure activities were not reprehensible. I agree here with R3: "Skating, at least according to Van Suchtelen (2001), symbolized recklessness and the transience of life for instance". R2 and R3 are totally right when they speak about art market and the mode of winter landscapes. I will add a short development in this direction.

Thank you also for your comments about minor corrections.

I will correct my paper and submit a second version.

[Figure]

[Figure]

**Fig. 1.**